

# Temperature responsiveness of soil carbon fractions, microbes, extracellular enzymes and CO$_2$ emission: mitigating role of texture

Waseem Hassan[1,2], Yu'e Li[2], Tahseen Saba[3], Jianshuang Wu[2], Safdar Bashir[4], Saqib Bashir[4], Mansour K. Gatasheh[5], Zeng-Hui Diao[6] and Zhongbing Chen[7]

[1] Landwirtschaftlich-Gärtnerischen, Humboldt-Universität zu Berlin, Berlin, Germany
[2] Institute of Environment and Sustainable Development in Agriculture/Laboratory for Agricultural Environment, Ministry of Agriculture and Rural Affairs, Chinese Academy of Agricultural Sciences, Beijing, China
[3] College of Forestry, Sichuan Agricultural University, Chengdu, Sichuan, China
[4] Department of Soil and Environmental Sciences, Ghazi University, Dera Ghazi Khan, Dera Ghazi Khan, Pakistan
[5] Department of Biochemistry, College of Science, King Saud University, Riyadh, Saudi Arabia
[6] School of Environmental Science and Engineering, Zhongkai University of Agriculture and Engineering, Guangzhou, China
[7] Department of Applied Ecology, Faculty of Environmental Sciences, Czech University of Life Sciences, Praha-Suchdol, Prague, Czech Republic

Corresponding authors
Waseem Hassan,
waseem.hassan@caas.cn
Zeng-Hui Diao,
zenghuid86@163.com

## ABSTRACT

The interaction of warming and soil texture on responsiveness of the key soil processes *i.e.* organic carbon (C) fractions, soil microbes, extracellular enzymes and CO$_2$ emissions remains largely unknown. Global warming raises the relevant question of how different soil processes will respond in near future, and what will be the likely regulatory role of texture? To bridge this gap, this work applied the laboratory incubation method to investigate the effects of temperature changes (10–50 °C) on dynamics of labile, recalcitrant and stable C fractions, soil microbes, microbial biomass, activities of extracellular enzymes and CO$_2$ emissions in sandy and clayey textured soils. The role of texture (sandy and clayey) in the mitigation of temperature effect was also investigated. The results revealed that the temperature sensitivity of C fractions and extracellular enzymes was in the order recalcitrant C fractions > stable C fractions > labile C fractions and oxidative enzymes > hydrolytic enzymes. While temperature sensitivity of soil microbes and biomass was in the order bacteria > actinomycetes > fungi ≈ microbial biomass C (MBC) > microbial biomass N (MBN) > microbial biomass N (MBP). Conversely, the temperature effect and sensitivity of all key soil processes including CO$_2$ emissions were significantly ($P < 0.05$) higher in sandy than clayey textured soil. Results confirmed that under the scenario of global warming and climate change, soils which are sandy in nature are more susceptible to temperature increase and prone to become the CO$_2$-C sources. It was revealed that clayey texture played an important role in mitigating and easing off the undue temperature influence, hence, the sensitivity of key soil processes.

# INTRODUCTION

The world's soils store substantially more carbon (C) than is present in the atmosphere (*Badgery et al., 2020*; *Paustian et al., 2019*). The estimated global soil C pool at one-meter depth is >1,500 GT and two-meter depth is >2,500 GT, which is 3.2 and four times the size of combined atmospheric and biotic C pool (*Zomer et al., 2017*). Being a gigantic pool, terrestrial C is receiving increasing attention both as a potentially large and uncertain source of $CO_2$ and also as a natural sink to reduce atmospheric $CO_2$ (*Badgery et al., 2020*; *Zomer et al., 2017*). It has been estimated that soils emit ≥11 times $CO_2$-C than fossil fuel combustion which is roughly about 68–100 $Pgy^{-1}$ (*Zhang & Zhou, 2018*). Conversely, even a 0.4% annual increase in soil C has the potential to significantly halt the yearly atmospheric $CO_2$ increase (*Amundson & Biardeau, 2019*). Therefore, it is of utmost importance to examine soil C and its divergent fractions, and their likely sensitivity and response towards temperature increase for future feedbacks and predictions.

Due to continuous movement in the soil systems, soil C is constantly disintegrating and changing into divergent pools (*Yang et al., 2021*; *Zomer et al., 2017*). The three major pools of soil C are recalcitrant C pool (RCP), labile C pool (LCP), and stable C pool (SCP) respectively (*Zhang & Zhou, 2018*). The LCP is composed of newly incorporated plant residues, amino acids, simple carbohydrates, root exudates, and simple C fractions (*Lian et al., 2018*). Whereas, RCP is made of detritus, decomposed plant and microbial byproducts, and C fractions *e.g.*, recalcitrant organic carbon (ROC) which is resistant to decomposition (*Zhang & Zhou, 2018*). Whereas, the total organic C (TOC) which is a heterogeneous mixture of diverse compounds (*e.g.*, residues, humin, humic acid, aromatic and hydrophobic compounds) with several hundred years of a mean age and accounts for 90% of stable fraction is also known as SCP (*Lian et al., 2018*). These C fractions are of utmost importance, owing to their direct and strong role in soil structure, C cycling and production and fluxes of $CO_2$ (*Badgery et al., 2020*; *Zomer et al., 2017*). The C fractions are extremely susceptible to abiotic variables, and multiple earlier studies have demonstrated that the future C balance of terrestrial ecosystems is highly dependent on the consequences of global warming (*Wang et al., 2016*; *Qi et al., 2016*; *Biswas et al., 2018*). *Qi et al. (2016)* observed a significant reduction in soil labile organic C fractions in response to warming, while *Karhu et al. (2010)* reported a decline in stable organic carbon fractions. Nonetheless, the temperature sensitivity of C fractions is a highly controversial and vague topic to date (*Davidson & Janssens, 2006*; *Sierra, Malghani & Loescher, 2017*). Therefore, it is the need of the day to quantify and establish the temperature sensitivity of C fractions of labile, recalcitrant, and stable pools.

Soil microbes *i.e.*, bacteria, fungi, and actinomycetes, owing to their vast metabolic diversity play diverse and critical roles in all-major biogeochemical cycles and ecosystem services (*Walker et al., 2018*; *Nottingham et al., 2019*). They also play a key role in

regulating the C decomposition, emission of $CO_2$, and overall C cycle of the ecosystem (*Qu et al., 2020*). Alike, soil enzymes are major components of biological processes which participate in all biochemical reactions (*Hassan et al., 2013a*). Soil enzymes play an important role in the biological catabolism, decomposition of organic matter and C cycling (*Hassan et al., 2013b*; *Aislabie & Deslippe, 2013*). They also perform catalysis of reactions that are necessary for the life processes of microorganisms (*Walker et al., 2018*; *Hassan et al., 2013b*). Soil microbial community and enzymes respond to changes in soil and environmental factors much faster than do other variables (*Nottingham et al., 2019*; *Aislabie & Deslippe, 2013*). Soil microbial community and enzymes are sensitive to a number of environmental factors, among them temperature is of utmost ascendancy (*Walker et al., 2018*). Under the scenario of global warming, it is indeed important to test the temperature sensitivity of the soil microbial communities (bacteria, fungi, and actinomycetes) and extracellular enzymes (oxidative and hydrolytic).

Temperature is rightly known as one of the primary bio-controller, because it influences soil reactions, biological processes and the inter-spheric gas exchange between the soil and atmosphere (*Thakur et al., 2017*; *Fang et al., 2016*). Due to its control over energy shifts, microbial communities, and extracellular enzyme activity, it regulates OM mineralization rates and storage, and hence the production of $CO_2$ in soils is also temperature-dependent (*Hassan, David & Abbas, 2014*; *Thakur et al., 2017*; *Walker et al., 2018*). Therefore, the temperature has a great influence on the ability of soils to act as a C sink or source (*Walker et al., 2018*; *Thakur et al., 2017*; *Fang et al., 2016*). For example, *Zhou et al. (2013)* found that a 6-year warming period enhanced the activities of β-glucosidase and N-acetylglucosaminidase, which were connected with changes in microbial biomass C. Under warming conditions, changes in the soil LOC fractions have been shown to drive changes in soil enzyme activity (*Zhou et al., 2013*; *Qi et al., 2016*). However, according to *Li et al. (2014)*, microbial responses to climate change may be influenced by soil properties.

The texture is one of the most important properties of soil because it determines characteristics and biophysical properties that shape and regulate the overall behavior and response of soils (*Fang et al., 2016*; *Ding et al., 2014*; *Hassan et al., 2013a*). The texture is associated with porosity, moisture, gaseous exchange, nutrient cycling, and substrate availability to microbiota along with other important functions and services in soils (*Oertel et al., 2016*; *Hobley et al., 2014*; *Hamarashid, Othman & Hussain, 2010*). Moreover, it also provides physical protection to soil microbiota, organic matter, and C from harsh climatic conditions *i.e.*, temperature anomalies (*Frøseth & Bleken, 2015*; *Hassan et al., 2013a*). Therefore, it affects the microbial and enzymatic activity, decomposition of organic matter, nutrients and C cycling, and eventually $CO_2$ production (*Ding et al., 2014*; *Feng, Plante & Six, 2013*). Soil texture can modulate the effects of temperature and climate change and thus production and emission of gases (*e.g.*, $CO_2$) through its strong influence on biochemical processes and C cycling and storage (*Zhang et al., 2015*; *Feng, Plante & Six, 2013*). The main reason for the strong influence of texture on key soil processes and activities is diverse and divergent characteristics of its relative particle's *i.e.*, fine and coarse (*Frøseth & Bleken, 2015*; *Hamarashid, Othman & Hussain, 2010*). The fine

particles (*i.e.*, clay) have large surface areas, numerous reactive sites, strong ligand exchange, and polyvalent cation bridges than coarse ones *i.e.*, sand (*Fang et al., 2016*; *Ding et al., 2014*; *Hassan et al., 2013a*). However, the interaction of warming and soil texture on responsiveness of the key soil processes remains largely unknown.

Therefore, it is indeed important to quantify the role of texture in regulating the temperature sensitivity of key soil processes for correct future inventories and feedbacks. We hypothesized that, warming would increase decomposition of recalcitrant and stable soil C pools *via* microbial activities and extracellular enzymes. These changes will be more pronounced in sandy textured soils while clayey soils will mitigate the effects of warming. Thus, the purpose of this study was (1) to determine the temperature influence and responsiveness of labile, recalcitrant, and stable C fractions, as well as $CO_2$ emission from divergent textured soils (2) quantify the effect of temperature on soil microbial counts (bacteria, fungi, and actinomycetes), microbial biomass, and extracellular enzymes (oxidative and hydrolytic) activities and their response towards temperature increase in divergent textured soils and (3) identify and establish the potential role of texture in climate change mitigation.

## MATERIAL AND METHODS

### Soil sampling

The study area has a moderately continental climate, the maximum and minimum mean annual temperatures are 14.03 °C and 6.72 °C and average annual precipitation is 24.97 mm. Soil samples (0–30 cm depth) were collected randomly using a hand auger from 10 points within the selected agricultural fields at Dahlem and Rhinluch, Berlin, Germany (52°27″ N and 13°18″ E) in April 2017. Winter wheat and maize was grown in rotation. Soils were Albic Luvisol and Arenosol with glacial till and periglacial sand parent materials. Samples (field fresh) were sieved (<2 mm) and separated into two subsamples. One part of the subsample was used to conduct the incubation experiment while the other was used for microbial and enzymatic analyses. The remaining soil was used for physicochemical and C fractional analyses after air-drying at room temperature (25 °C) for 7 days by using methods as described by *Hassan, David & Abbas (2014)*. The basic physicochemical properties of experimental soil are given in Table 1A.

### Experimental layout

For incubation, 400 g dry soil was incubated in 1,000 ml glass jars under different temperature and moisture regimes for 84 days in a randomized block design. Soil samples were wetted to maintain 60% of water holding capacity (WHC) and equilibrated overnight at 4 °C, before being placed in incubators. The five treatments in triplicate were developed and expressed as T1 (10 °C), T2 (20 °C), T3 (30 °C), T4 (40 °C), and T5 (50 °C). To keep the soils at their prescribed WHC, moisture loss in the jars was determined after every 2 days by weighing the jars and the water loss was replenished with distilled water throughout the incubation (*Elliott et al., 1994*). Soil samples were collected from each jar after incubation, for the determination of labile, recalcitrant and stable C fractions, microbial community and enzymes.

**Table 1A Physico-chemical properties of experimental soils.**

| Properties | Textural class | |
| --- | --- | --- |
| | Sandy loam | Clay loam |
| Sand (%) | 539.3 | 232.4 |
| Silt (%) | 251.2 | 286.3 |
| Clay (%) | 209.5 | 481.3 |
| pH (1:2.5) | 7.11 | 7.24 |
| EC ($\mu$S cm$^{-1}$) | 149.8 | 169.5 |
| CEC (C mol$_c$ kg$^{-1}$) | 7.32 | 8.35 |
| Bulk density (g cm$^3$) | 1.58 | 1.28 |
| Available N (mg kg$^{-1}$) | 5.61 | 6.91 |
| Available P (mg kg$^{-1}$) | 4.63 | 6.49 |
| Available K (mg kg$^{-1}$) | 170.5 | 201.8 |
| Total P (g kg$^{-1}$) | 0.18 | 0.32 |
| Total N (g kg$^{-1}$) | 0.37 | 0.53 |

## Quantification of Carbon dioxide ($CO_2$) emission

The emission of $CO_2$ from the incubated soil (as described above) was estimated by the alkali trap method as described by *Witkamp (1966)*. The evolved $CO_2$ was trapped in 25 ml of 0.1 M KOH. After exposure, the KOH solution was removed, and any carbonate formed precipitated with saturated $BaCl_2$ to form $BaCO_3$; the remaining KOH was then titrated with an equivalent strength of HCl using phenolphthalein as an indicator. A jar without soil, containing the same amount of KOH, was run simultaneously as a blank. The evolved $CO_2$ was measured at 7, 21, 42, 63 and 84 days during incubation. The evolved and cumulative $CO_2$ was calculated, by using the method of *Hassan, David & Abbas (2014)*.

## Determination of soil C fractions

### Total organic carbon

Total soil carbon of the soil before and after the experiment was determined by potassium dichromate ($K_2Cr_2O_7$) oxidation at 170–180 °C followed by titration with 0.5 mol L$^{-1}$ $FeSO_4$ (*Walkley & Black 1934*).

### Light fraction of organic carbon

Light fraction of organic carbon was measured by wet oxidation ($K_2Cr_2O_7$). Fleetingly, 50 ml NaI solution (1.70 g cm$^{-3}$ density) along with soil sample (25 g) was placed into a centrifuge tube and was shaken (200 rpm) for 15 min. The floating material was extracted in triplicate and transferred to a filter paper and rinsed every time with $CaCl_2$ (0.01 M) and distilled water, and then dried (60 °C) for 48 h (*Gregorich & Ellert, 1993*).

### Readily mineralizable carbon

Readily mineralizable carbon was estimated after extraction with $K_2SO_4$ (0.5 M) followed by wet digestion with dichromate. Briefly, soil (10 g), after precipitating the $Fe^{2+}$ with 1 ml

of $FeCl_3$ (2.5% solution) and 4 ml of 6 N NaOH, was extracted with 40 ml of $K_2SO_4$ (0.5 M) after shaking for 1 h at a rotary shaker. After allowing the precipitate to settle down (4 °C) clear supernatant (aliquots) were titrated with $FeH_8N_2O_8S_2$ (0.04 N) by using 2 to 3 drops of diphenylamine (DPA) indicator after wet digestion with $H_2CrO_4$ (*Mishra et al., 1997*).

### Dissolved organic carbon

For dissolved organic carbon fresh soil (10 g) was extracted with the 2.0 M KCl (1:4 soil/water) after shaking (250 rpm) the soil samples for 30 min. The supernatant was then centrifuged (15,000 rpm) for 10 min and filtered (0.45 μm cellulose ester filters) and analyzed at a TOC (Multi N/C 2100, Germany) analyzer (*Zsolnay, 2003*).

### Particulate organic carbon

The particulate organic carbon was quantified after dispersing the soil sample (10 g) with 30 ml of hexametaphosphate (5 g $l^{-1}$) in a reciprocating shaker (90 rpm) for 18 h. The soil suspension was transferred into another clean and empty container under a continuous flow of distilled water over a sieve (53-μm) to ascertain the separation. The remaining soil on the sieve was dried at 55–60 °C for 48 h after shifting to a glass dish, and ground to powder with a ball mill, and measured (wet digestion) for POC by using $K_2Cr_2O_7$ (*Cambardella & Elliott, 1992*).

### Reducing sugar carbon

The content of reducing sugar carbon was determined using a phenol reagent. One ml of soil extract was mixed with 1 ml of the phenol solution (5% w/v in distilled water), then 5 ml of 18.4 M $H_2SO_4$ (1.84 d) was added under continuous shaking. The mixture was left for 10 min, thereafter, incubated in a water bath at 25 °C for 20 min and the absorbance was read colorimetrically with a standard curve of glucose at 490 nm by following *Badalucco et al. (1992)* with slight modification.

### Easily oxidizable carbon

For easily oxidizable carbon soil (3 g) was centrifuged (2,000 rpm) for 5 min along with 25 ml of $KMnO_4$ (333 mM) and the absorbance of the supernatant and standards was read spectrometrically at 565 nm. Likewise, the blank samples (no soil + standard) were also analyzed in each run. The change in the concentration of $KMnO_4$ was used to assess the amount of C oxidized by assuming that 1 mM $MnO_4$ is consumed for the oxidation of 0.75 mM or 9 g of C (*Blair, Lefory & Lise, 1995*).

### Recalcitrant organic carbon

The recalcitrant organic carbon was determined by the acid hydrolysis (18 h) of soil (1 g) with HCl (6 M). The repeated evaporation and filtration were done in order to remove the HCl and separate the soluble materials. The residue was washed with de-ionized water (20 ml) and dried at 55 °C. After drying, the residue was ground and passed through a screen (180 mm), and combusted to $CO_2$ (*Paul, Morris & Bohm, 2001*).

## Analysis of soil microbial colony counts and biomass

The total number of bacteria, fungus, and actinomycetes was determined using the dilution plate count technique on nutritional agar, as described previously by *Hassan et al. (2013b)*. The dilution plate technique is based on the assumption that each colony is created by a single cell, referred to as a colony-forming unit (CFU). In a flask containing 90 ml distilled water and glass beads, 10 g of fresh soil was added (0.5 mm). For 30 min, the flask was shaken at 28 °C and 180 rpm. A total of 0.1 ml of the suspension was put to a small tube containing 0.9 ml distilled water after shaking. The tube was gently shaken and used to perform the remaining dilutions. To count bacteria, dilutions of $10^{-1}$–$10^{-8}$ were utilized. Conversely, a range of $10^{-1}$–$10^{-6}$ was used for the determination of fungi and actinomycete. Each dilution was repeated three times In an incubator, the plates were incubated at 28 °C (301.15 K). Bacteria, fungi, and actinomycetes were identified 4, 5, and 7 days after plating, respectively (*Hassan et al., 2013b*). The chloroform fumigation-extraction method was used to determine the microbial biomass, *i.e.*, MBC, MBN, and MBP. For this purpose, 10 g fresh soil was fumigated with alcohol-free chloroform for 24 h at a temperature of 25 °C in a desiccator. The soils (fumigated and non-fumigated) were then extracted for an hour using a horizontal shaker (200 rpm), filtered with Whatman No. 40 filter paper, and finally spectrophotometrically measured and computed (*Hassan et al., 2013b*).

## Examination of enzymes activity

### Phenoloxidase and peroxidase activity

The phenoloxidase and peroxidase activity was measured by incubating (25 °C) the soil (0.5 g), in a shaking environment (100 rpm), with acetate buffer (3 ml) and 2 ml of 10 mM L-3,4-dihydroxyphenylalanine (L-DOPA), followed by centrifugation for 10 min at 5 °C. For peroxidase, an addition of 0.3% $H_2O_2$ (0.2 ml), just before incubation, was made. Then the absorbance of the dopachrome (reaction product) was read at 475 nm spectrophotometrically and activity of both enzymes was expressed as µmol dopachrome $g^{-1}$ $h^{-1}$ (*Dick, 2011*).

### Catalase activity

The catalase activity was measured by titrating residual $H_2O_2$ in the filtrate with $KMnO_4$ (0.1 N), after mixing the soil (1 g) with 3% $H_2O_2$ (1 ml) and $H_2SO_4$ (5 ml) after shaking (20 min), followed by filtration. The activity was expressed as µmol $H_2O_2$ $g^{-1}$ $h^{-1}$ (*Roberge, 1978*).

### Invertase activity

The activity of invertase was determined by incubating (24 h at 37 °C) the soil (5 g) with sucrose solution (15 ml), phosphate buffer (35.6 g $Na_2HPO_4$ + 700 ml distilled $H_2O$ + adjust pH to 5.5 with HCl + volume to 1 l) and toluene (4–5 drops), followed by filtration. The activity (color density) was measured spectrophotometrically at 508 nm, after mixing the filtrate (1 ml) with 0.5% $C_7H_4N_2O_7$ (2 ml), heating (5 min) in a boiling water bath and colling (3 min) it down under running water and making the final volume to 25 ml with deionized $H_2O$ and described as µmol glucose $g^{-1}$ $h^{-1}$ (*Dick, 2011*).

### β-glucosidase activity

The β-glucosidase activity was estimated by incubating and treating (1 h at 37 °C) the soil (1 g) with 0.25 ml toluene, 0.25 mn p-nitrophenol phosphate (p-NPP), 4 ml MUB (Modified universal buffer), 1 ml of glucoside, 1 ml of $CaCl_2$ (0.5 M) and 4 ml of 0.1 M THAM (Tris-hydroxymethyl-aminomethane) solution. After filtration (Whatman No. 2V) the activity was determined through spectrophotometer at 400 nm and described as µmol p-nitrophenol $g^{-1}$ $h^{-1}$ (*Eivazi & Tabatabai, 1988*).

### Cellulase activity

The cellulase activity was measured after incubating (24 h, 50 °C), centrifuging (2,500$g$, 10 min) and treating the soil (10 g) with 5 ml acetate buffer (11.2 M, pH 5.5), carboxymethyl cellulose sodium (7 g) and cellulose substrate (0.7%). After filtration the activity was determined through spectrophotometer at 690 nm and described as µmol glucose $g^{-1}$ $h^{-1}$ (*Schinner & Von Mersi, 1990*).

## Statistical analysis

The statistical software Statistix 8.1 (Statistix, Tallahassee, FL, USA), and Excel 2016 were used for data analysis. Parametric statistics of ANOVA analysis was carried out to estimate the effect of temperature on the soil microbes, microbial biomass, enzymes, C fractions and $CO_2$ production and emissions under divergent textures. Mean separations were achieved by using the least significant difference (LSD) test at $P < 0.05$. Data presented are means ± standard deviation (SD) of three replicates of each treatment. Correlation coefficients ($R^2$) between soil C fractions of labile, recalcitrant, and stable pools, $CO_2$ emissions and cumulative $CO_2$, microbial community, microbial biomass, oxidative and hydrolytic enzymes and temperature were developed by using the same software.

## RESULTS

### Labile C fractions

The response and decomposition of labile C fractions viz LFOC, DOC, RMC, RSC, POC, and EOC under a range of elevated temperature (T1–T5) regimes in sandy and clayey soil is presented in Fig. 1. The response and decomposition of labile C fractions increased significantly ($P < 0.05$) with the increase in the temperature (per 10 °C rise). However, the temperature response and decomposition of labile C fractions were significantly ($P < 0.05$) higher in sandy than the clayey soil. Therefore, in sandy soil, the maximum increase in the LFOC (2.92-fold), DOC (3.34-fold), RMC (4.07-fold), RSC (4.54-fold), POC (3.51-fold), and EOC (4.02-fold) was observed at the highest temperature *i.e.*, T5 compared to lowest temperature (T1). Conversely, in clayey soil, maximum increase in the LFOC (2.41-fold), DOC (2.05-fold), RMC (3.17-fold), RSC (2.98-fold), POC (2.71-fold), and EOC (3.03-fold) was observed at the T4 compared to lowest temperature (T1). Whereas, the minimum sensitivity and decomposition of labile C fractions were observed at the lowest temperature *i.e.*, T1. Furthermore, owing to higher temperature impact and decomposition, the sandy soil exhibited significantly lower labile C fractions *i.e.*, LFOC
(1.14-fold), DOC (1.17-fold), RMC (1.14-fold), RSC (1.15-fold), POC (1.16-fold), and EOC (1.17-fold) compared to the clayey soil. Mainly the effect of temperature on the labile C fractions in the sandy soil was in the order T4 > T5 > T3 > T2 > T1. Conversely, the influence of temperature on the labile C fractions in the clayey soil was in the order T5 > T4 > T3 > T2 > T1.

## Recalcitrant C fractions

The response and decomposition of recalcitrant C fraction viz ROC under a range of elevated temperature (T1–T5) regime in sandy and clayey soil is illustrated in Fig. 2. The response and decomposition of ROC enhanced markedly ($P < 0.05$) with the temperature increase (per 10 °C rise) in both textured soils. Unlike the labile C fractions, the increase in temperature caused a significant and continuous increase in the response and decomposition of ROC under both textured soils. Therefore, the maximum increase in the ROC (3.16-fold and 3.72-fold) was observed at the highest temperature *i.e.*, T5 in sandy and clayey soil respectively compared to lowest temperature (T1). Whereas, the minimum response and decomposition of ROC were observed at the lowest temperature *i.e.*, T1. Due to the higher temperature effect and decomposition, the decrease in the ROC was higher (1.15-fold) in sandy compared to clayey soil. In general, the effect of temperature on the ROC in both soils was in the order T5 > T4 > T3 > T2 > T1 endorsing, the fact that ROC likely has higher sensitivity to temperature (T1–T5) increase than the labile C fractions.

## Stable C fractions

The response and decomposition of stable C fractions viz TOC under a range of elevated temperature (T1–T5) regime in sandy and clayey soil is presented in Fig. 3. The response and decomposition of TOC enhanced greatly ($P < 0.05$) with the temperature increase (per 10 °C rise) in both textured soils. Unlike the labile and stable C fractions the increase in temperature significantly ($P < 0.05$) decreased the TOC. Therefore, the maximum decrease in the TOC (3.89-fold and 3.60-fold) was observed at the highest temperature (T5) in both sandy and clayey soil respectively compared to lowest temperature (T1). Highlighting that TOC has a strong ($P < 0.05$) antagonistic association to the temperature increase (T1–T5). Whereas, the minimum response and decomposition of TOC were observed at the lowest temperature *i.e.*, T1. Owing to the higher temperature effect and decomposition, the decrease in the TOC was higher (1.14-fold) in sandy compared to the clayey soil. In general, the effect of temperature on the TOC decomposition and sensitivity was in the order T5 > T4 > T3 > T2 > T1.

## Microbial community

The response of soil microbes *i.e.*, bacteria, fungi, and actinomycetes under a range of elevated temperature (T1–T5) regimes in sandy and clayey soils is exhibited in Fig. 4. The response of the soil microbes increased significantly ($P < 0.05$) with the temperature increases (per 10 °C rise) in both textured soils. However, the temperature sensitivity and response of the soil microbes were significantly ($P < 0.05$) variable. The bacteria

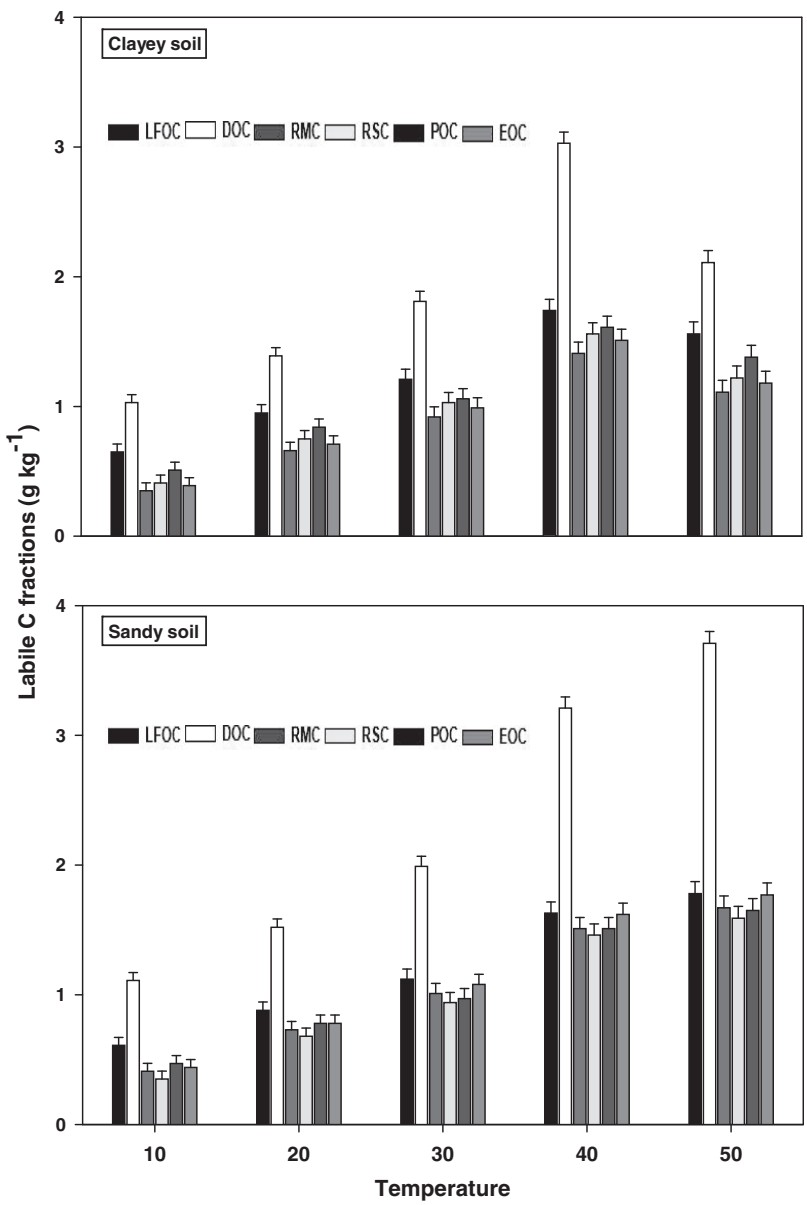

**Figure 1 Effect of temperature on labile C fractions under sandy and clayey texture.** LFOC, light fraction of organic carbon; DOC, Dissolve organic carbon; RMC, readily mineralizable carbon; RSC, reducing sugar carbon; POC, Particulate organic carbon; EOC, easily oxidizable carbon. Vertical bars represent means ± SD ($n$ = 3). ANOVA significant at $P \leq 0.05$.

showed the maximum temperature response (2.22-fold and 2.57-fold) at the highest temperature (*i.e.*, T5) in sandy and clayey soils correspondingly compared to lowest temperature (T1). Conversely, the maximum increase in the response of actinomycetes (2.01-fold and 2.52-fold) and fungi (1.64-fold and 1.73-fold) were found at T4 and T3 in sandy and clayey soils respectively compared to lowest temperature (T1). Indicating the fact that among soil microbes the temperature sensitivity order is bacteria > actinomycetes > fungi. Whereas, the minimum sensitivity and response of soil microbes

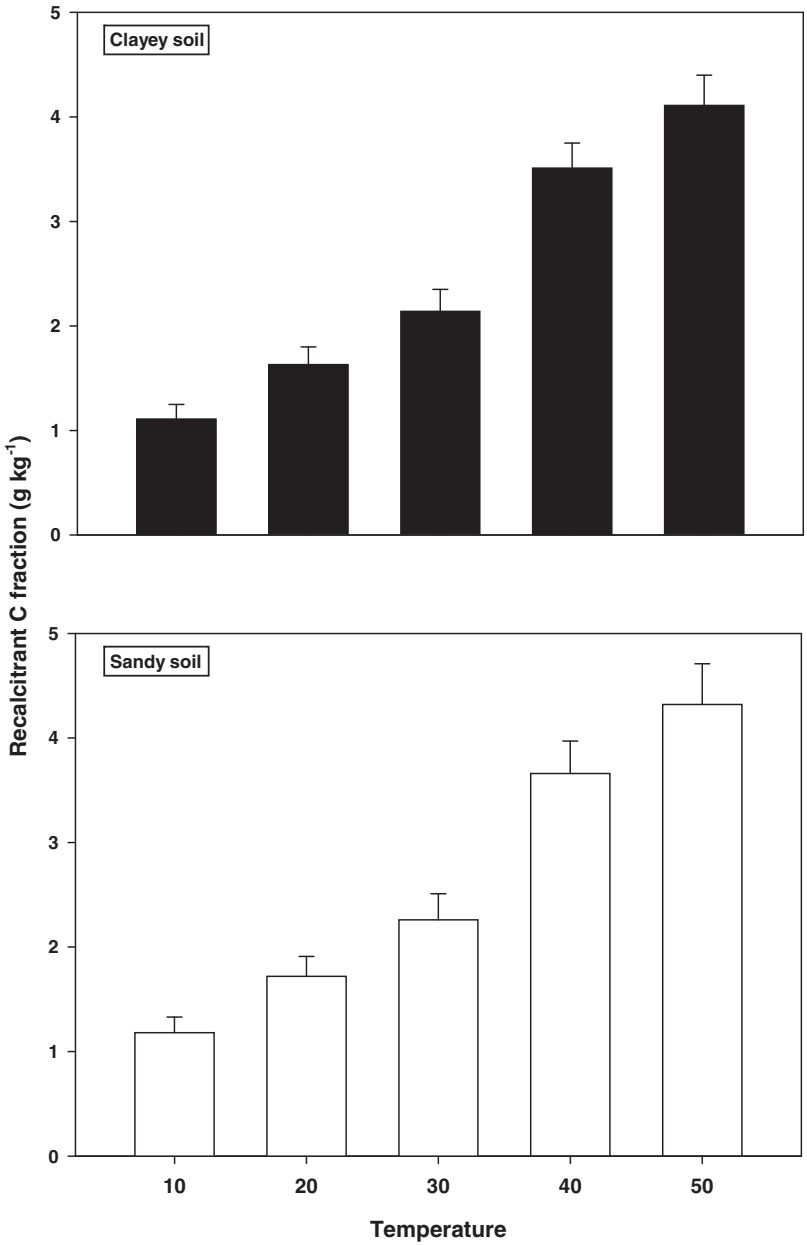

**Figure 2 Effect of temperature on recalcitrant C fraction under sandy and clayey texture.** ROC, recalcitrant organic carbon. Vertical bars represent means ± SD ($n$ = 3). ANOVA significant at $P \leq$ 0.05.

were observed at the lowest temperature *i.e.*, T1. Besides, owing to higher temperature effect and sensitivity, the sandy soil showed a markedly lower soil microbes count *i.e.*, bacteria (1.34-fold), fungi (1.12-fold), and actinomycetes (1.14-fold) compared to clayey soil. In general, the effect of temperature on the bacterial counts was in the order T5 > T4 > T3 > T2 > T1. Whereas the temperature sensitivity of actinomycetes and fungi were in an order of T4 > T5 > T3 > T2 > T1 and T3 > T4 > T5 > T2 > T1 respectively.

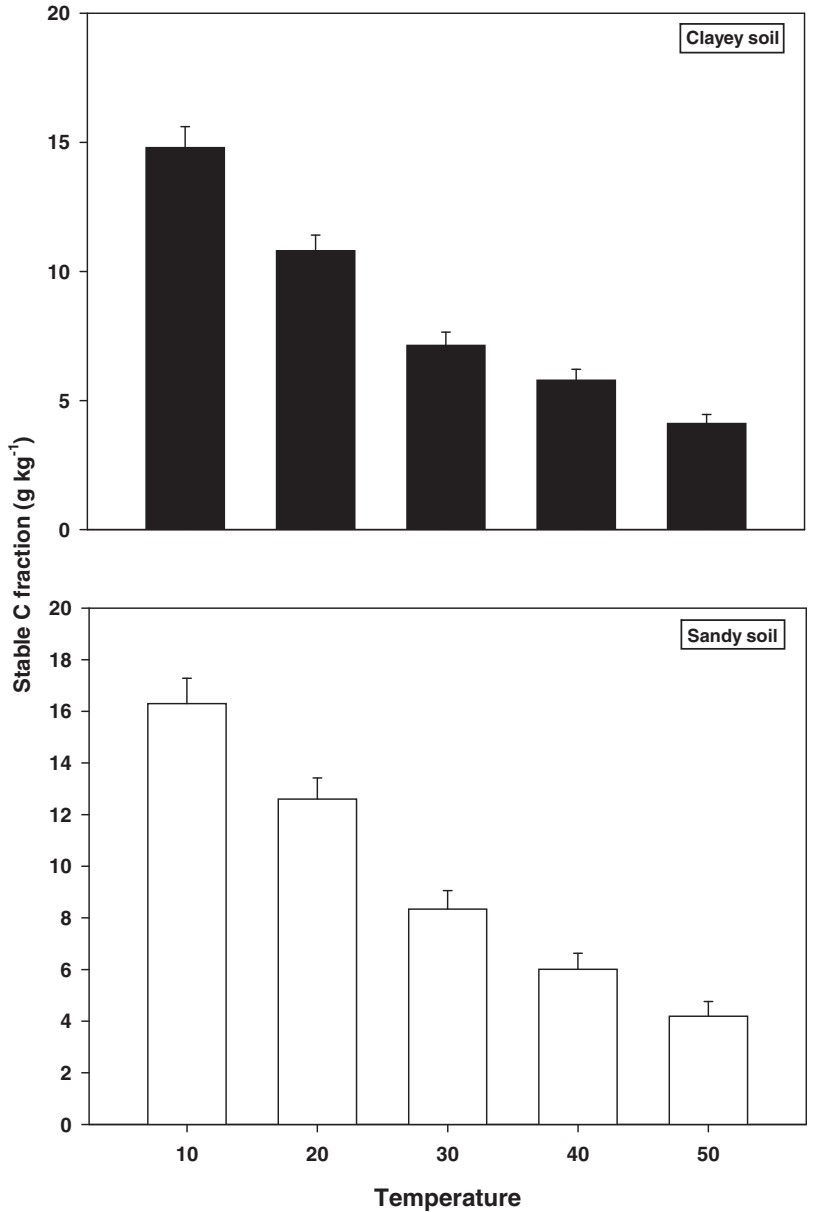

**Figure 3 Effect of temperature on stable C fraction under sandy and clayey texture.** TOC, total organic carbon. Vertical bars represent means ± SD ($n = 3$). ANOVA significant at $P \leq 0.05$.

## Microbial biomass

The response of microbial biomass *i.e.*, MBC, MBN, and MBP under a range of elevated temperature (T1–T5) regimes in sandy and clayey soils is presented in Fig. 5. The response of microbial biomass increased significantly ($P < 0.05$) with the temperature surge (per 10 °C rise) in both textured soils. However, like soil microbes colony counts, the temperature sensitivity of microbial biomass was also significantly ($P < 0.05$) variable. The MBC exhibited the maximum temperature sensitivity and increase (1.97-fold and 2.21-fold) at the highest temperature (*i.e.*, T5) in sandy and clayey soils correspondingly

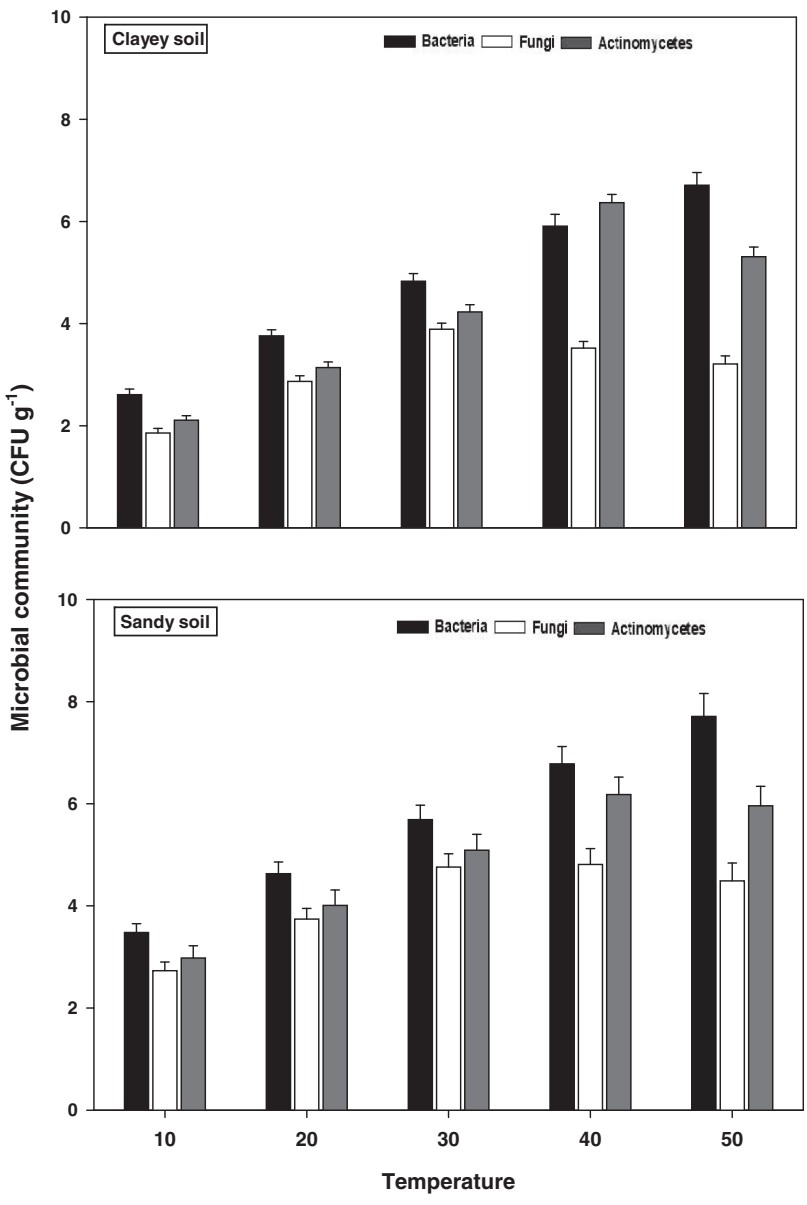

**Figure 4 Effect of temperature on microbial community under sandy and clayey texture.** Units: bacteria, CFU $\times 10^6$ g$^{-1}$; fungi, CFU $\times 10^4$ g$^{-1}$; Actinomycetes, CFU $\times 10^5$ g$^{-1}$. Vertical bars represent means ± SD ($n$ = 3). ANOVA significant at $P \leq 0.05$.

compared to lowest temperature (T1). On the contrary, the maximum increase in the MBN (2.11-fold and 2.22-fold) and MBP (1.84-fold and 2.31-fold) were found at T4 and T3 in sandy and clayey soils respectively compared to lowest temperature (T1). Indicating the fact that among microbial biomass the temperature sensitivity order is MBC > MBN > MBP. Whereas, the minimum sensitivity and response of microbial biomass were observed at the lowest temperature *i.e.*, T1. Moreover, due to the higher temperature effect and sensitivity the sandy soil exhibited a significantly lower microbial biomass *i.e.*, MBC (1.23-fold), MBN (1.29-fold), and MBP (1.43-fold) compared to the clayey soil. The temperature sensitivity order for MBC was T5 > T4 > T3 > T2 > T1. Whereas the

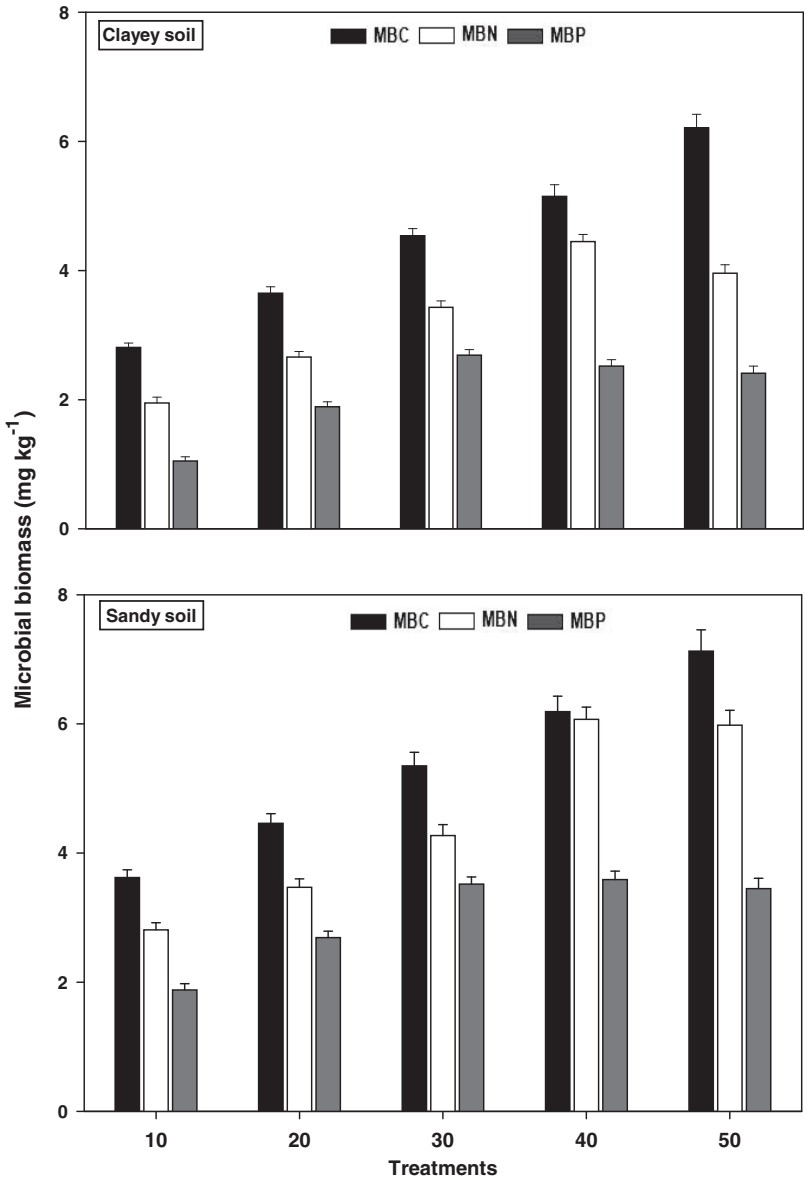

**Figure 5 Effect of temperature on microbial biomass under sandy and clayey texture.** MBC, microbial biomass carbon, MBN, microbial biomass nitrogen; MBP, microbial biomass phosphorous. Vertical bars represent means ± SD ($n$ = 3). ANOVA significant at $P \leq 0.05$.

temperature sensitivity order for MBN and MBP was T4 > T5 > T3 > T2 > T1 and T3 > T4 > T5 > T2 > T1 respectively.

## Oxidative enzymes

The response and activity of oxidative enzymes viz PO, PEO, and CAT under a range of elevated temperature (T1–T5) regimes in sandy and clayey soil are shown in Fig. 6. However, unlike the hydrolytic enzymes, a significant ($P < 0.05$) and continuous increase in the activity of oxidative enzymes was observed with the increase in the temperature

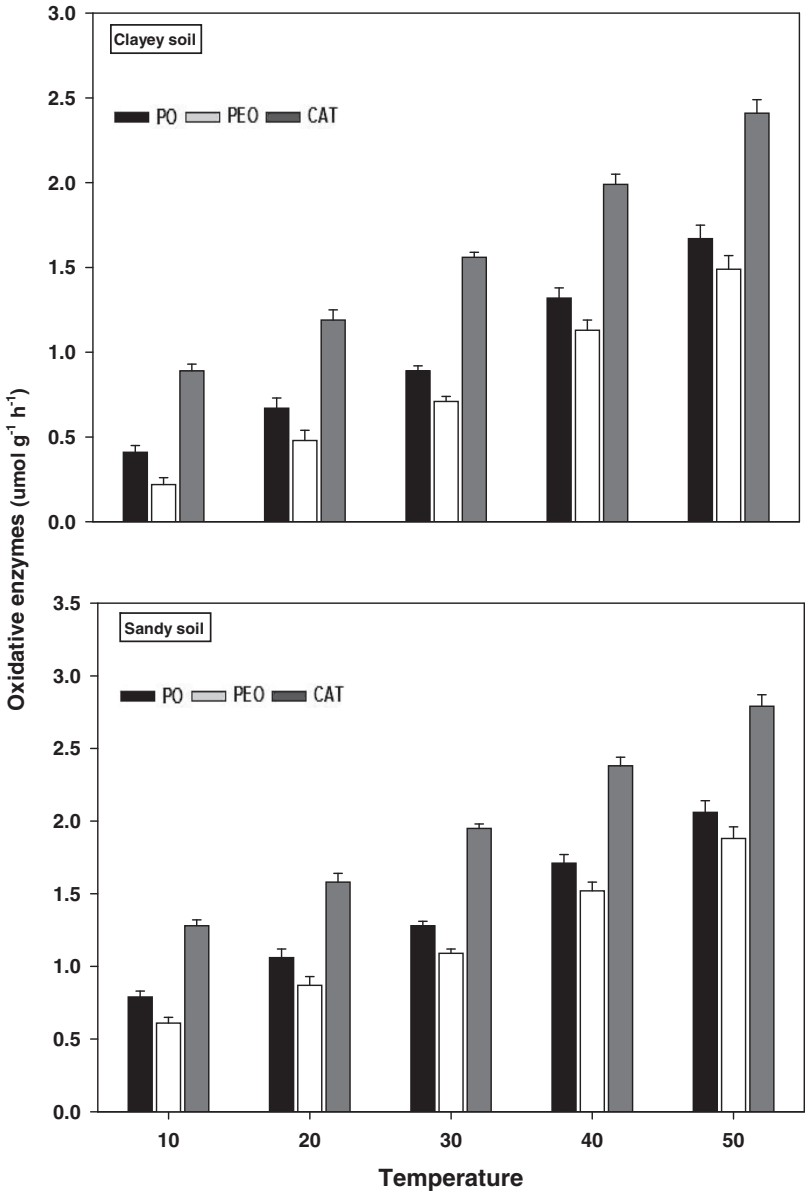

**Figure 6 Effect of temperature on oxidative enzymes activity under sandy and clayey texture.** PO, phenol oxidase; PEO, peroxidase; CAT, catalase. Units: PO and PEO, µmol dopachrome $g^{-1}$ $h^{-1}$; CAT, µmol $H_2O_2$ $g^{-1}$ $h^{-1}$. Vertical bars represent means ± SD ($n$ = 3). ANOVA significant at $P \leq 0.05$.

(per 10 °C rise). As a result, the maximum increase in the activity of PO (2.61-fold and 4.07-fold), PEO (3.08-fold and 6.77-fold), and CAT (2.18-fold and 2.71-fold) were found at the highest temperature *i.e.*, T5 in sandy and clayey soils respectively compared to lowest temperature (T1). Whereas minimum response and activity of oxidative enzymes were observed at the lowest temperature *i.e.*, T1. Establishing the fact that oxidative enzymes have decidedly higher responsiveness to temperature increase than the hydrolytic enzymes. Furthermore, owing to the higher temperature effect, the sandy soil showed a markedly lower activity and values of oxidative enzymes *i.e.*, PO (1.69-fold), PEO

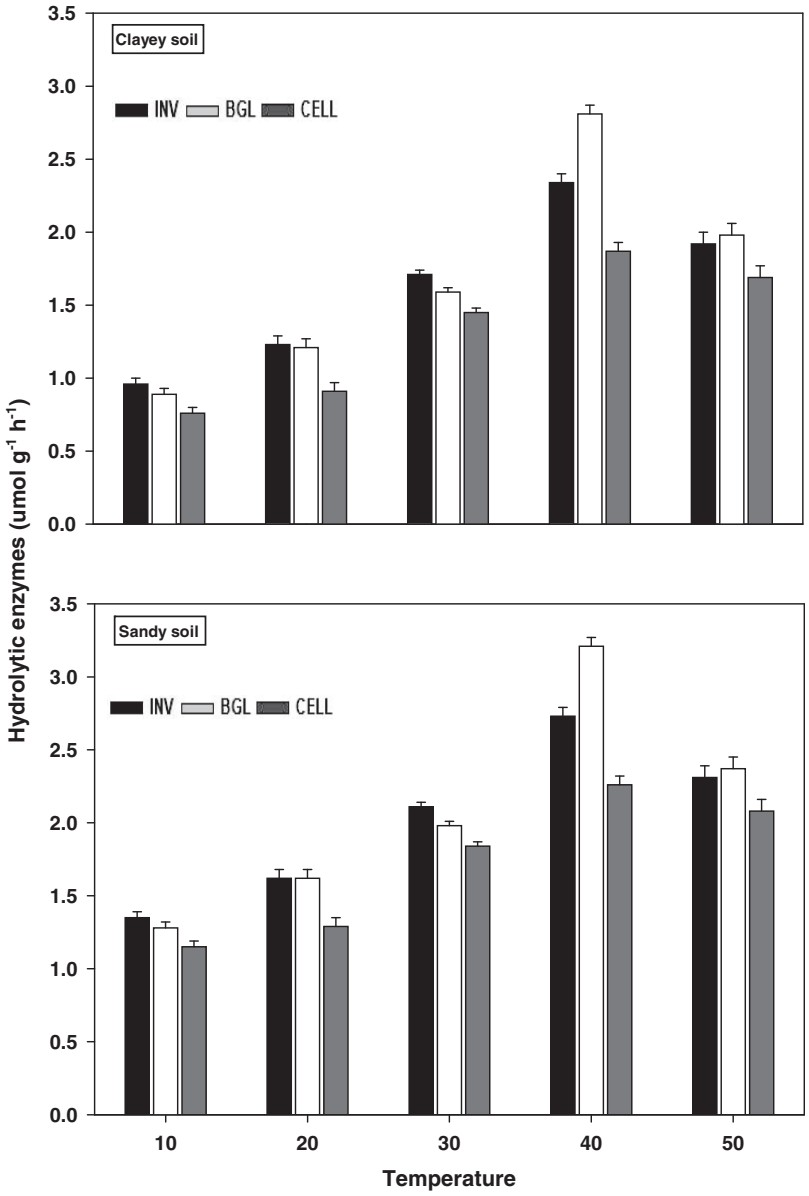

**Figure 7 Effect of temperature on hydrolytic enzymes activity under sandy and clayey texture.** INV, Invertase; BGL, β-glucosidase; CELL, cellulose. Units: INV, μmol glucose $g^{-1}$ $h^{-1}$; BGL, μmol *p*-nitro-phenol $g^{-1}$ $h^{-1}$; CELL, μmol glucose $g^{-1}$ $h^{-1}$. Vertical bars represent means ± SD ($n$ = 3). ANOVA significant at $P \leq 0.05$.

(1.48-fold), and CAT (1.24-fold) compared to the clayey soil. The overall effect of temperature on the sensitivity and activity of oxidative enzymes was in the order T5 > T4 > T3 > T2 > T1.

## Hydrolytic enzymes

The response and activity of hydrolytic enzymes viz INV, BGL, and CELL under a range of elevated temperature (T1–T5) regimes in sandy and clayey soils are depicted in Fig. 7. Generally, an increasing trend was observed in the activity of hydrolytic enzymes under

elevated temperature (per 10 °C rise). However, unlike the oxidative enzymes, the maximum increase in the activity of INV (1.71-fold and 2.01-fold), BGL (1.85-fold and 2.22-fold), and CELL (1.81-fold and 2.23-fold) were found at T4 in sandy and clayey soils respectively compared to lowest temperature (T1). After that, an abrupt decrease in the activity of hydrolytic enzymes was examined at the highest temperature *i.e.*, T5 compared to lowest temperature (T1). Whereas, the minimum activity of hydrolytic enzymes was observed at the lowest temperature (T1). Additionally, due to the higher temperature effect, the sandy soil depicted a significantly lower activity and values of hydrolytic enzymes *i.e.*, INV (1.25-fold), BGL (1.23-fold), and CELL (1.29-fold) compared to the clayey soil. The overall effect of temperature on the response and activity of hydrolytic enzymes was in the order T4 > T5 > T3 > T2 > T1.

### Emissions and cumulative $CO_2$

The response and changes in the emissions and cumulative $CO_2$ under a range of elevated temperature (T1–T5) regimes in sandy and clayey soils and a graphical summary of methodology are illustrated in Figs. 8 and 9. Overall, an increasing trend was found in the emissions and cumulative $CO_2$ for each 10 °C rise in temperature. However, the temperature responsiveness and changes in emissions and cumulative $CO_2$ were significantly ($P < 0.05$) higher in sandy than the clayey soil. Therefore, in sandy soil, the maximum increase in the emissions (1.84-fold) and cumulative $CO_2$ (1.81-fold) were observed at the highest temperature (T5) compared to lowest temperature (T1). Conversely, in clayey soil, higher emissions (1.45-fold) and cumulative $CO_2$ (1.36-fold) were observed at the T4 compared to lowest temperature (T1). After that, in clayey soil, a decrease in the response and thus values of emissions and cumulative $CO_2$ were examined at the highest temperature *i.e.*, T5. Whereas, the minimum sensitivity and changes in emission and cumulative $CO_2$ were observed at the lowest temperature (*i.e.* T1). Furthermore, unlike other key soil processes, the sandy soil showed greater increase in the emissions (1.22-fold) and cumulative $CO_2$ (1.23-fold) owing to higher temperature effect, decomposition rate and changes than the clayey soil. Underscoring the fact that $CO_2$ production and emissions and cumulative $CO_2$ have positive feedback with the augmentation in the temperature, and sandy soil are more vulnerable than the clayey ones. Mainly the effect of temperature on the sensitivity and emissions and cumulative $CO_2$ in the sandy soil was in the order T5 > T4 > T3 > T2 > T1. Conversely, the influence of temperature on the sensitivity and emission and cumulative $CO_2$ in the clayey soil was in the order T4 > T5 > T3 > T2 >.

### Regression analysis

Regression analysis showed that C fractions of labile, recalcitrant, and stable pools, $CO_2$ fluxes, and cumulative $CO_2$ correlated well with the temperature in both sandy and clayey soils (Table 1B). Nonetheless, overall, temperature accounted for 93% and 79% variability in the C fractions of labile, recalcitrant, and stable pools in the sandy and clayey soils correspondingly (Table 1B). Whereas, temperature accounted for 91% and 94% variability in the $CO_2$ fluxes and cumulative $CO_2$ in the sandy soil. Conversely, temperature described

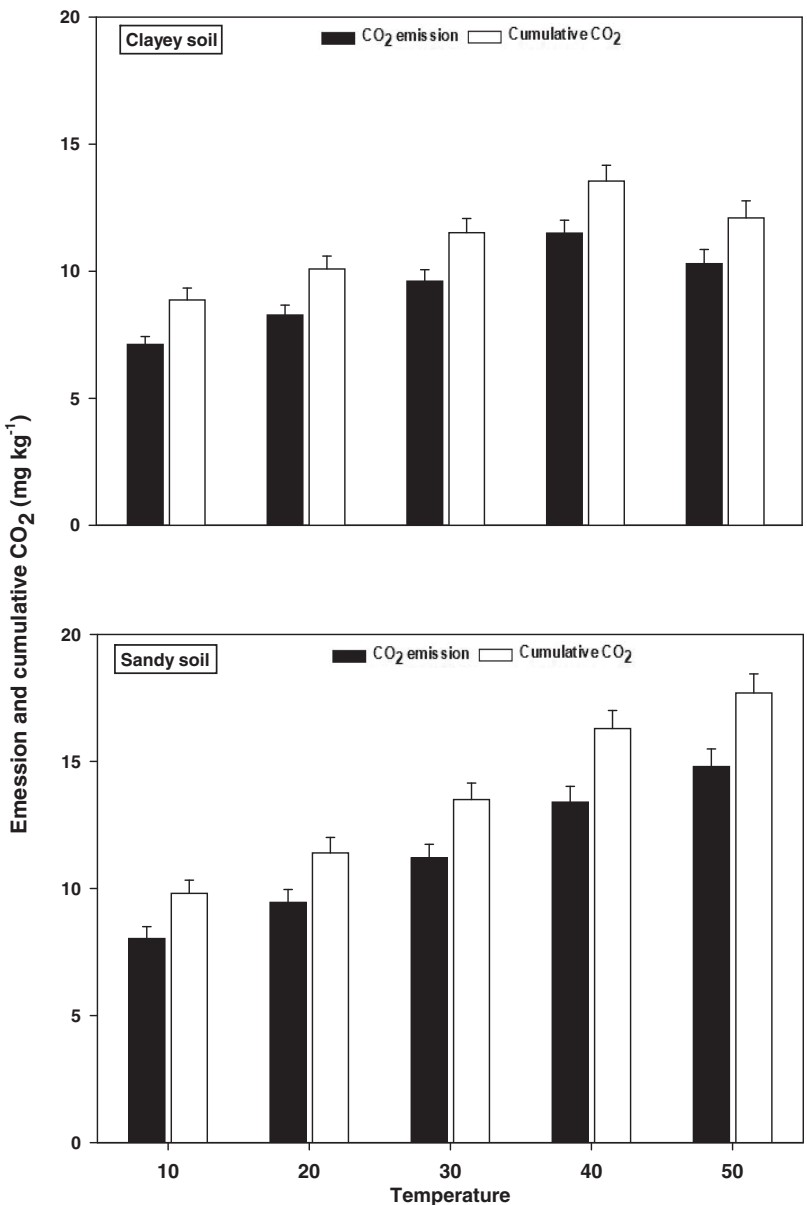

**Figure 8 Effect of temperature on $CO_2$ emissions and cumulative $CO_2$ under sandy and clayey texture.** Unit: $CO_2$ emission, mg kg$^{-1}$ h$^{-1}$; Cumulative $CO_2$, mg kg$^{-1}$. Vertical bars represent means $\pm$ SD ($n = 3$). ANOVA significant at $P \leq 0.05$.

for 78% and 75% alterability in the $CO_2$ fluxes and cumulative $CO_2$ in the clayey soil (Table 1B). Furthermore, temperature accounted variability was significantly higher for C fractions of recalcitrant and stable pools compared to labile pools in both sandy and clayey soils (Table 1B). Regression analysis showed that overall, temperature accounted for 85% and 72% variability in the microbial community in the sandy and clayey soils (Table 2A). The temperature accounted variability was significantly higher for bacteria ($R^2 = 0.97$ and 0.91) than actinomycetes ($R^2 = 0.92$ and 0.81) and fungi ($R^2 = 0.68$ and 0.46) in both sandy and clayey soils (Table 2A). Moreover, the temperature described

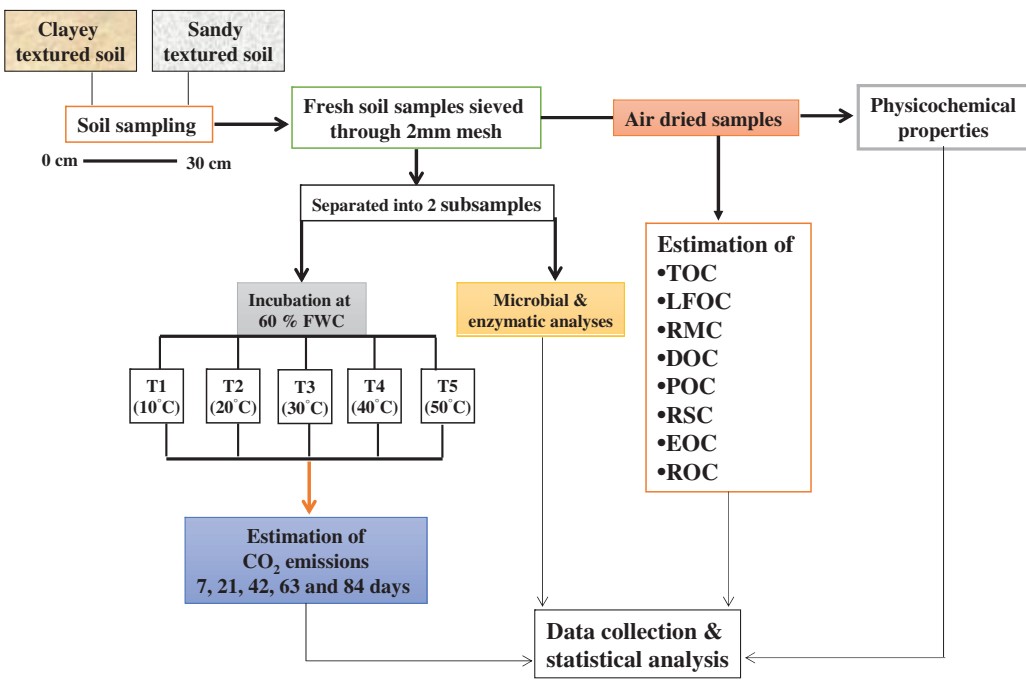

**Figure 9 Methods graphical representation.**

**Table 1B Correlation coefficient ($R^2$) between C fractions, $CO_2$ emissions, cumulative $CO_2$ and temperature.**

| Parameters | $R^2$ (Sandy soil) | $R^2$ (Clayey soil) |
|---|---|---|
| LFOC | 0.93** | 0.85** |
| DOC | 0.94** | 0.62* |
| RMC | 0.92** | 0.75* |
| RSC | 0.91** | 0.76* |
| POC | 0.92** | 0.71* |
| EOC | 0.93** | 0.84** |
| [a]Total $R^2$ | 0.92** | 0.75* |
| ROC | 0.97** | 0.91** |
| TOC | 0.96** | 0.89** |
| [b]Total $R^2$ | 0.97** | 0.90* |
| $CO_2$ emission | 0.91** | 0.78* |
| Cumulative $CO_2$ | 0.94** | 0.75* |

**Notes:**
[a] Total $R^2$, Correlation coefficient from all labile C fractions.
[b] Total $R^2$, Correlation coefficient from all microbial biomass.
* Significant at $P < 0.05$.
** Significant at $P < 0.01$.

alterability was markedly higher in sandy soil than clayey soil (Table 2A). Whereas, temperature designated for 88% and 78% variability in the microbial biomass in the sandy and clayey soils (Table 2A). The temperature accounted alterability was significantly higher for MBC ($R^2$ = 0.98 and 0.92) than MBN ($R^2$ = 0.91 and 0.80) and MBP ($R^2$ = 0.75
**Table 2A  Correlation coefficient (R$^2$) between microbial community and biomass and temperature.**

| Parameters | R$^2$ (Sandy soil) | R$^2$ (Clayey soil) |
|---|---|---|
| Bacteria | 0.97** | 0.91** |
| Fungi | 0.68* | 0.46* |
| Actinomycetes | 0.92* | 0.81* |
| [a]Total R$^2$ | 0.85** | 0.72* |
| MBC | 0.98** | 0.92** |
| MBN | 0.91** | 0.80* |
| MBP | 0.75* | 0.63* |
| [b]Total R$^2$ | 0.88** | 0.78* |

Notes:
[a] Total R$^2$, Correlation coefficient from all microbial community.
[b] Total R$^2$, Correlation coefficient from all microbial biomass.
* Significant at $P < 0.05$.
** Significant at $P < 0.01$.

**Table 2B  Correlation coefficient (R$^2$) between oxidative and hydrolytic enzymes and temperature.**

| Parameters | R$^2$ (Sandy soil) | R$^2$ (Clayey soil) |
|---|---|---|
| PO | 0.96** | 0.88** |
| PEO | 0.93** | 0.86** |
| CAT | 0.91** | 0.84** |
| [a]Total R$^2$ | 0.93** | 0.86** |
| INV | 0.76* | 0.69* |
| BGL | 0.64* | 0.58* |
| CELL | 0.81* | 0.73* |
| [b]Total R$^2$ | 0.73* | 0.66* |

Notes:
[a] Total R$^2$, Correlation coefficient from all oxidative enzymes.
[b] Total R$^2$, Correlation coefficient from all hydrolytic enzymes.
* Significant at $P < 0.05$.
** Significant at $P < 0.01$.

and 0.63) in both sandy and clayey soils (Table 2A). Moreover, the temperature designated changeability was markedly higher in sandy soil than clayey soil (Table 2A). Regression analysis showed that overall, temperature accounted for 93% and 86% variability in the oxidative enzymes in the sandy and clayey soils (Table 2B). Whereas, temperature accounted for 73% and 66% variability in the hydrolytic enzymes in the sandy and clayey soils (Table 2B). The temperature described alterability was significantly higher for oxidative enzymes than hydrolytic enzymes. Moreover, the temperature accounted variability was markedly higher in sandy soil than clayey soil (Table 2B).

## DISCUSSION

The responsiveness and decomposition of labile C fractions increased significantly ($P < 0.05$) with the temperature increase (per 10 °C) in both textured soils (Fig. 1). *Yang et al. (2021)* and *Qi et al. (2016)* assessed that temperature increase significantly alters the fractions of soil labile organic C (RSC, MBC, DOC, and POC) by increasing their response and rate of decomposition. However, response to the temperature and

decomposition of labile C fractions were significantly ($P < 0.05$) higher in sandy soil than the clayey soil (Fig. 1). Temperature accounted variability for labile C fractions was significantly higher in sandy soil than clayey soil (Table 1B). *Wankhede et al. (2020)*, *Rittl et al. (2020)*, *Takriti et al. (2018)*, *Ghosh et al. (2016)*, *Frøseth & Bleken (2015)* and *Hobley et al. (2014)* found that temperature impacts on labile C fractions was higher in coarse (sandy) than fine (clayey) soils owing to low physical protection, small specific areas, fewer reactive sites, and weak ligand exchange bridges, where soil C could be sorbed and protected. In present study, unlike the labile C fractions, the increase in temperature (T1–T5) caused a significant ($P < 0.05$) and continuous increase in the sensitivity and decomposition of recalcitrant (ROC) and stable (TOC) C fractions (Figs. 2 and 3) in both soils. Whereas, the temperature impacts and thus decomposition of ROC (1.15-fold) and TOC (1.14-fold) were significantly higher in sandy soil at T5 *i.e.* 50 °C (Figs. 2 and 3). *Wankhede et al. (2020)* and *Zheng et al. (2019)* examined that in sandy (coarse) soils the temperature response of recalcitrant and stable C fractions was much higher due to weak physical protection, fewer cations bridges, unstable moisture availability, and their low storing ability. The response of C fractions to temperature was in the order recalcitrant C fractions > stable C fractions > labile C fractions (Figs. 1–3). In both sandy and clayey soils, temperature accounted variability was significantly higher for C fractions of recalcitrant and stable pools compared to labile pools (Table 1B). *Zhang & Zhou (2018)* and *Dai et al. (2017)* found that recalcitrant and stable C fractions have decidedly extra sensitivity than the labile fractions to the temperature increase (5 °C to 30 °C) in divergent coarse and fine textured Chinese soils. The results of current study also endorsed the fact that recalcitrant and stable C fractions have a higher sensitivity to temperature (T1–T5) increase than the labile C fractions (Figs. 1–3). Higher $R^2$ were found for recalcitrant and stable C fractions than labile ones in both sandy and clayey soils (Table 1B). *Biswas et al. (2018)*, *Lian et al. (2018)*, *Fang et al. (2016)* and *Nguyen et al. (2010)* confirmed that recalcitrant and stable C fractions have higher responses to temperature than the labile C fractions in coarse and fine textured soils.

The response of soil microbial counts and microbial biomass increased markedly ($P < 0.05$) with the temperature increase (per 10 °C rise) in both sandy and clayey textured soils. However, the temperature responses of microbial colony counts *i.e.*, bacteria (1.34-fold), fungi (1.12-fold), and actinomycetes (1.14-fold) and biomass *i.e.*, MBC (1.23-fold), MBN (1.29-fold), and MBP (1.43-fold) were higher in sandy soil (Figs. 4 and 5). Overall, in sandy soil, temperature accounted for significantly higher variability in microbial population (85%) and microbial biomass (88%) than in clayey soil (Table 2A). *Qu et al. (2020)*, *Nottingham et al. (2019)*, *Hutchins et al. (2019)*, *Zhang et al. (2016)*, *Fang et al. (2016)*, and *Hassan et al. (2013a)* examined that microbial counts (bacteria, fungi, and actinomycetes) and biomass (MBC, MBN, and MBP) had higher sensitivity to temperature increase and their sensitivity increased many folds in coarse (sandy) soils owing to less favorable conditions, predation, desiccation, and substrate availability. The results further, revealed that temperature sensitivity and response of soil microbes colony counts and biomass were significantly variable (Figs. 4 and 5). *Cavicchioli et al. (2019)*, *Zhang et al. (2016)* and *Fang et al. (2016)* also examined variations in the activity,

behavior, and response of microbial community and biomass towards experimental warming and temperature increase. The temperature response of soil microbes colony counts and biomass were in the order bacteria > actinomycetes > fungi and MBC > MBN > MBP (Table 2A). Therefore, the maximum activity and response of bacteria, actinomycetes and fungi and MBC, MBN, and MBP were observed at temperatures T5, T4 and T3 in both soils respectively (Figs. 4 and 5). *Zheng et al. (2019)*, *Dubey et al. (2019)*, *Walker et al. (2018)*, and *Zhang et al. (2016)* found a strong association between temperature increase and responses of soil microbes colony counts and biomass and stated that temperature sensitivity of bacteria and MBC is much higher followed by actinomycetes and fungi and MBN and MBP in diverse textured soils (coarse and fine). The temperature accounted variability was significantly higher for bacteria ($R^2 = 0.97$ and $R^2 = 0.91$) than actinomycetes and fungi in both sandy and clayey soils (Table 2A). *Romero-Olivares, Alisson & Trescedar (2017)*, *Zhang et al. (2016)*, *García-Palacios et al. (2015)*, and *Wang et al. (2014)* also stated that among microbes and biomass, bacterial community and MBC have decidedly higher sensitivity, contrarily, fungi are less sensitive to changes in temperature owing to the chitinous cell walls that make them highly resilient. The temperature accounted variability was significantly higher for MBC ($R^2 = 0.98$ and $R^2 = 0.92$) than MBN and MBP in both sandy and clayey soils (Table 2A). *Melillo et al. (2017)*, and *Crowther et al. (2016)* also found a significant association between the increase in temperature (warming), temperature sensitivity, and reduction in the microbial biomass and stated that temperature sensitivity of MBC is markedly higher.

The extracellular enzymes (*i.e.*, oxidative and hydrolytic) response and activity increased significantly ($P < 0.05$) with the temperature increase (per 10 °C rise) in both textured soils. However, the temperature response of oxidative enzymes *i.e.*, PO (1.69-fold), PEO (1.48-fold), and CAT (1.24-fold) and hydrolytic enzymes *i.e.*, INV (1.25-fold), BGL (1.23-fold), and CELL (1.29-fold) were markedly higher in sandy soil (Figs. 6 and 7). The temperature accounted variability for oxidative and hydrolytic enzymes was markedly higher in sandy soil than clayey soil (Table 2B). *Wankhede et al. (2020)*, *Cavicchioli et al. (2019)*, *Zheng et al. (2019)*, *Thakur et al. (2017)*, and *Fang et al. (2016)* assessed a significant increase in the sensitivity and response of extracellular enzymes *i.e.*, oxidative and hydrolytic with the temperature increase and stated that temperature sensitivity increases strongly in sandy (coarse) soils due to less favorable conditions, unstable moisture, and substrate availability. The results of present study further revealed that the temperature sensitivity of extracellular enzymes was in the order oxidative enzymes > hydrolytic enzymes (Figs. 6 and 7). The temperature accounted variability was significantly higher for oxidative enzymes (93% and 86%) than hydrolytic enzymes (73% and 66%) in both sandy and clayey soils (Table 2B). Establishing the fact that oxidative enzymes have higher temperature sensitivity than the hydrolytic enzymes in both sandy and clayey soils. *Meng et al. (2020)*, *Tang et al. (2019)*, *Walker et al. (2018)*, *Allison et al. (2018)*, *Cheng et al. (2017)*, and *Fang et al. (2016)* examined a strong synergistic association between extracellular enzymes sensitivity and temperature and revealed that oxidative enzymes (*e.g.*, PO, PEO, and CAT) have decidedly higher

temperature sensitivity than the hydrolytic (*e.g.*, DEH, URE, INV, BGL, and PHP) enzymes.

Overall, an increasing trend was found in the emissions and cumulative $CO_2$ under elevated *i.e.*, each 10 °C rise in temperature in both sandy and clayey soils (Fig. 8). However, the temperature effect and changes in emissions and cumulative $CO_2$ were significantly ($P < 0.05$) higher in sandy than the clayey soil. Temperature accounted for 91% and 94% variability in the $CO_2$ emissions and cumulative $CO_2$ in the sandy soil (Table 1B). *Sánchez-Cañete, Barron-Gaford & Chorover (2018)*, *Zomer et al. (2017)*, *Ekwurzel et al. (2017)*, *Fang et al. (2016)* and *Frøseth & Bleken (2015)* examined a significant increase in the emissions and cumulative $CO_2$ with the temperature rise and stated that coarse *i.e.*, sandy soils have much higher temperature effect thus emissions and cumulative $CO_2$ than the fine (clayey or silty) soils. Therefore, in sandy soil, the maximum increase in the CO emissions (1.84-fold) and cumulative $CO_2$ (1.81-fold) was observed at the highest temperature (T5). Furthermore, the sandy soil showed significantly ($P < 0.05$) higher $CO_2$ emissions (1.22-fold) and cumulative $CO_2$ (1.23-fold) owing to higher temperature effect, response and decomposition rate than the clayey soil (Fig. 8). Significantly higher correlation coefficients were observed between $CO_2$ emissions ($R^2 = 0.91$) and cumulative $CO_2$ ($R^2 = 0.94$) and temperature in sandy soil than clayey soil (Table 1B). *Wachiye et al. (2019)*, *Badagliacca et al. (2017)*, *Frøseth & Bleken (2015)* and *Ding et al. (2014)* found a significantly higher responsiveness of $CO_2$ emissions and cumulative $CO_2$ to temperature in the sandy soils and established that this was due to high rate of C decomposition, low humification, and small specific areas in sandy soils, where soil C could be sorbed, secured and stored. Underscoring the fact that $CO_2$ production and emissions and cumulative $CO_2$ have a strong synergistic association with the temperature augmentation, and sandy soils have much higher temperature sensitivity and vulnerability to become $CO_2$-C sources than the clayey ones (Fig. 8 and Table 1B). Temperature accounted for significantly lower variability for the $CO_2$ fluxes (78%) and cumulative $CO_2$ (75%) in clayey soil compared to sandy soil (Tables 2A and 2B). *Oertel et al. (2016)*, *Frøseth & Bleken (2015)*, *Zhang et al. (2015)* and *Six & Paustian (2014)* inspected that the sandy soils are more sensitive to temperature increase and the main reason of sandy/coarse soils to foster higher $CO_2$ production and emissions and cumulative $CO_2$ is high decomposition rate, low humification and availability of C sorption, and attachment sites.

## CONCLUSION

The study concluded that the temperature sensitivity of soil C fractions, microbial colony counts, microbial biomass, extracellular enzymes, and $CO_2$ fluxes increased with the upsurge in temperature. However, the recalcitrant and stable C fractions have decidedly higher responses than labile C fractions. Alike, among microbes, microbial biomass, and extracellular enzymes, bacteria, MBC, and oxidative enzymes (PO, PEO, and CAT) have markedly higher sensitivity. It was concluded that the temperature effect and variability for all measured key soil processes along with $CO_2$ fluxes were markedly higher in sandy textured soil. Conversely, clayey texture performed a significant role in the

mitigation of undue temperature influence, hence, the sensitivity of key soil processes and $CO_2$ fluxes. The study also suggests between sandy and clayey textured soils, the soils which are sandy in nature under the scenario of global warming, are more vulnerable to become $CO_2$-C sources therefore must be managed and treated wisely. Furthermore, in future research and models instead of generalizing effects of global warming, temperature sensitivity of individual key soil processes must also be considered carefully. The findings of the study will be helpful in alleviating the controversy of the temperature sensitivity of key soil processes in sandy and clayey soils. And enabling the scientists and environmentalists to formulate measures and devise recommendations to reduce the excessive increase in $CO_2$-C fluxes from divergent textured soils.

## LIST OF ABBREVIATIONS

| | |
|---|---|
| **TOC** | total organic C |
| **MBC** | microbial biomass C |
| **MBN** | microbial biomass N |
| **MBP** | microbial biomass P |
| **RCP** | recalcitrant C pool |
| **LCP** | labile C pool |
| **SCP** | stable C pool |
| **ROC** | recalcitrant organic carbon |
| **WHC** | water holding capacity |
| **LFOC** | light fraction of organic carbon |
| **RMC** | readily mineralizable carbon |
| **DOC** | dissolved organic carbon |
| **POC** | particulate organic carbon |
| **EOC** | easily oxidizable carbon |
| **ROC** | recalcitrant organic carbon |
| **CFU** | colony-forming unit |

### Funding

The present study was conducted with the support of the China Postdoctoral Council and the Institute of Environment and Sustainable Development in Agriculture (Grant No. NNSFC 42007073 and MARA, PRC 13210352). This work was also supported by the Special project in key areas of Guangdong Province Ordinary Universities (No. 2020ZDZX1003), the Guangdong Provincial Special Fund for Modern Agriculture Industry Technology Innovation Teams (No. 2019KJ140), the Key Real R&D Program of Guangdong Province (2020B1111350002 and 2020B0202080002), and the National Natural Science Foundation of China (No. 21407155). Support was also provided by the Researchers Supporting Project number (RSP-2021/393), King Saud University, Riyadh,

Saudi Arabia. The funders had no role in study design, data collection and analysis, decision to publish, or preparation of the manuscript.

## Grant Disclosures
The following grant information was disclosed by the authors:
China Postdoctoral Council and Institute of Environment and Sustainable Development in Agriculture: 42007073 and 13210352.
Guangdong Province Ordinary Universities: 2020ZDZX1003.
Modern Agriculture Industry Technology Innovation Teams: 2019KJ140.
Key Real R&D Program of Guangdong Province: 2020B1111350002 and 2020B0202080002.
National Natural Science Foundation of China: 21407155.
Researchers Supporting Project number: RSP-2021/393.

## Competing Interests
The authors declare that they have no competing interests.

## Author Contributions
- Waseem Hassan conceived and designed the experiments, performed the experiments, prepared figures and/or tables, and approved the final draft.
- Yue Li conceived and designed the experiments, authored or reviewed drafts of the paper, and approved the final draft.
- Tahseen Saba analyzed the data, prepared figures and/or tables, and approved the final draft.
- Jianshuang Wu conceived and designed the experiments, authored or reviewed drafts of the paper, and approved the final draft.
- Safdar Bashir performed the experiments, authored or reviewed drafts of the paper, and approved the final draft.
- Saqib Bashir performed the experiments, authored or reviewed drafts of the paper, and approved the final draft.
- Mansour K. Gatasheh performed the experiments, authored or reviewed drafts of the paper, and approved the final draft.
- Zeng-Hui Diao conceived and designed the experiments, authored or reviewed drafts of the paper, and approved the final draft.
- Zhongbing Chen conceived and designed the experiments, authored or reviewed drafts of the paper, and approved the final draft.

## Data Availability
The raw data is available in the Supplemental File.

## Supplemental Information
Supplemental information for this article can be found online at http://dx.doi.org/10.7717/peerj.13151#supplemental-information.

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
