# Peer review of "Temperature responsiveness of soil carbon fractions, microbes, extracellular enzymes and CO2 emission: mitigating role of texture"

_PeerJ, doi:10.7717/peerj.13151_

## Round 0.1 · original submission · Major Revisions

An interesting topic, which is also relevant to the journal and possibly of wider interest to scientific community. However, it needs substantial revision - both language/writing style, and content-wise/experimental methods and data. Specifically care should be taken on those aspects while preparing the revised manuscript, as suggested by reviewers.

Reviewer 1 ·

Basic reporting

In this clearly written article, the authors explore the effect of warming on microbial processes in soil, including microbial biomass, community structure, and soil carbon storage. The article has professional structure, and the raw data provided is clear.

Areas for improvement:
It would help to include a note in the raw data saying each line is a replicate. It was unclear what the three numbers per category are otherwise.

The authors could state hypotheses in the introduction and then evaluate them in the Discussion.

Please say more about how you selected the research site.

A brief literature search found this article (Koven et al. 2017) which might help put soil temperature range in context of other ecosystems: https://escholarship.org/content/qt7q68v57x/qt7q68v57x.pdf

Experimental design

The research question is clear and relevant, finding the effect of warming on soil microbial properties. Adding in the effect of soil type is also relevant, since broader modeling efforts may need to consider soil type when scaling findings to larger areas. Methods are described in detail.

Areas for improvement:

L127 what plants were growing in these soils? Knowing some of the soil history would put the results in context.

Statistical Analysis: A linear (or in some cases nonlinear) regression would make more sense for these results than ANOVA.

Validity of the findings

Conclusions are well stated and tie back to the original research question about the effect of warming on soil microbial properties. Appreciate the quantitative detail such as listing fold-change for each microbial group and soil property. The research is methodical and the variety of factors measured in the soil sample is valuable.

Areas for improvement:

Here or in the Introduction: how was the temperature range decided? Testing 10degC to 50degC is helpful in seeing a broad range of response, but one would expect the effect of climate warming to be just a few degrees C, right? Do you expect the broad scale patterns you found translate to the narrow scale of climate effects?

Adding in some background about the environment where the soils were sampled could help with this – what is the seasonal max and min temperature there?

L390 instead of “increased strongly”, please quantify this such as % increase in decomposition rate per 10degC
L404 When mentioning fold-change, this is over what temperature range?
L413 Here and elsewhere (L434, etc.), please add a transition so the reader knows you are referring to your own results, rather than those of another study.
L442 Please clarify that one R2 is for sandy soil, and the other is for clayey soil
L451 For these studies cited, what ecosystem where they studying?

Fig 2. Why would temperature increase recalcitrant C? It seems that temperature would decrease this. Also, there is nearly a fivefold increase in recalcitrant C which is surprising.

Fig 3 and elsewhere: Please add in the figure captions that this is after X days incubation.

Fig 4 and elsewhere: It would be easier to compare these graphs if the Y axis ranges were the same (0-10 in this case).

Please consider reworking some of the figures so that sandy and clayey soil is in the same panel, so it is easy to compare the two soil types. The other factors could then be separated out, such as having Emission and Cumulative CO2 in separate panels in Figure 8.

Additional comments

Enjoyed reading this article, as it points to the question of how soil C might be a feedback on warming, and considers variation by soil texture. This is definitely an unresolved question with much need for research. Good luck in preparing this article for publication.

Reviewer 2 ·

Basic reporting

This manuscript is admirable in that it measures many important components of soil, however unfortunately I found it difficult to follow. Although many components were measured, the relationships between these components are not conceptually integrated, so the organization of the manuscript is disjointed.

There are also many confusing sentences. For example (lines 13-14): C factions, microbes/biomass, and extracellular enzymes are not soil processes, but elements part of the soil environment than can influence key processes. I would recommend revising most sentences in the manuscript to increase clarity for the reader. Additionally, this manuscript has many strange word choices. For example (line 44): what does “incessant” movement in the soil systems mean here? This was very distracting when reading the manuscript.

The manuscript included relevant literature, however oftentimes what was actually measured by this study does not fit with the citation that was used (e.g., studies on temperature sensitivity).

For the most part, the figures, tables, and raw data are clear.
- Table 1A: What are the units here for sand/silt/clay?

Clear hypotheses are not stated in the manuscript.

Experimental design

This manuscript is within the aims and scope of this journal. While research questions and a knowledge gap is stated, the research questions are very broad and the experiment does not meet the stated goals. Some specific comments:

- Line 125: Can you justify storage at 4c for the microbial and enzymatic analysis? Were these processed right away? Usually these samples are stored at -20C or -80C if stored for more than a couple of days (e.g., German et al., 2011 Soil Biology & Biochemistry).

- Line 133: I am assuming these jars were kept open? Please specify. If kept close, the headspace seems too small. Were these jars kept in the dark?

- I am not familiar with the specific CO2 or C fraction methods used here, but it seems fine. How often was CO2 measured throughout the 84-days?

- Line 188: It is a bit misleading to say soil microbial community analysis when only colony counts were done. Please change the title of this section and corresponding text throughout the manuscript to reflect this. Usually the microbial community would be DNA, RNA, or PLFA type of analysis. It’s okay to do colony counts, but I wouldn’t really consider that microbial community analysis (more like an estimate of the potential relative density of the bacteria, fungi, and actinomycetes).

- Line 204: Please break this section into two paragraphs—one describing the colorimetric approach (phenoloxidase and peroxidase) and the other describing the fluoroumentric approach (B-glucosidase and cellulose)

Validity of the findings

I have some serious concerns about the conclusions drawn from this work:

- Temperature sensitivity is not measured in this manuscript despite interpretation of temperature sensitivity throughout the Results, Discussion, and Conclusion. Here, you are comparing responses at different temperatures, but that is different from measuring temperature sensitivity. Usually temperature sensitivity is reported using Q10 (which is not a great metric in my opinion) or using parameter estimates for temperature sensitivity from a model (e.g., Macromolecular Rate Theory, Arrhenius, or Square root model). Your analyses and overall interpretation in your manuscript would be much improved if you used one of these approaches to compare the temperature sensitivity rather than just comparing responses between different temperatures.

- The regression approach used to compare and interpret the temperature responses is also misleading. As it is, you are only interpreting the linear relationships between your measured variables and temperature and using that to make claims about the relative importance of temperature on each factor. However, some of these relationships are clearly nonlinear (example: fungi). That does not mean the importance of temperature on fungal growth is less important than bacteria, only that the relationship is less linear. Some other kind of statistical analysis (maybe comparing sum of squares?) would be more appropriate if you want to make these types of claims.

- Lines 292-293: Please clarify that this measurement is only of potential community growth in media (very different from soil conditions).

- Lines 510-512: This is too bold of a statement to make for only measuring two soils.

Reviewer 3 ·

Basic reporting

The authors have tried to evaluate and investigate the effect of increasing temperature on Soil composed of the microbial community, enzymes, C fractions of labile, recalcitrant, and stable pools. The authors have related this study to the current global warming issues. Though the study was interesting, there are many issues in the manuscript that have to be addressed. The main comments are (a) sentence making, (b) unnecessary using words to create a complex sentence, (c) grammatical errors and tenses, and (d) references missing. Therefore, it is requested that the authors address the comments and suggestions below to improve the quality of the manuscript. At present, I would like to reject the manuscript in its current form and suggest major revisions.

Experimental design

1. Check for grammatical errors in the abstract.
2. Use the conjunction word “and” in the proper place. Lots of and is seen in the abstract. Too much and in a sentence changes the meaning. Check and rewrite
3. Explain the difference between microbes and biomass for the sentence “microbes and their biomass”.
4. What is MBC, MBN, and MBP that has been mentioned in the abstract? If carbon is mentioned as C, then the above term also has to be expanded and explained. Also, mention the same in the introduction section. “Microbial biomass carbon (MBC), Microbial biomass nitrogen (MBN), and Microbial biomass phosphorous (MBP)
5. Change is 3.2 times and four times to “3.2 and 4 fold times”.
6. Mention what is C in the introduction section as mentioned in Abstract “Carbon (C).”
7. Provide more literature for Carbon and divergent fractions in Soil for better understanding. Mention the role of these fractions and why it is important to study the effect of global warming on these fractions. However, authors have mentioned that this is a vague topic to date, include some literature that has related studies to the current study.
8. Mention some literature regarding the phenoloxidase, peroxidase, invertase, β-glucosidase and cellulase role in Soil.
9. Is it humic acid or humin acid?
10. Mention what OM is? “Organic matter” and again place the word “and” in the necessary place.
11. Break long sentences into two sentences for the readers to understand the message conveyed.

Validity of the findings

1. It is not necessary to mention soil samples again and again in the soil sampling section.
2. The reference for “Decomposition of Leaf Litter in Relation to Environment, Microflora, and Microbial Respiration” illustrates the CO2 capture method from incubated Soil is not included in the reference section. Please include and check for other missing references in the materials and methods section.
3. Check for grammatical mistakes and tenses in the materials and methods section.
4. It is necessary to mention the concentration of H2SO4 used during the analysis of reducing sugar. Recheck the methodology for sugar analysis. Badalucco et al. 1992 report the analysis carried out in OD of 485 nm, but the current study has carried out the work in 490 nm. In addition, the mixture was incubated in a water bath at 28oC for 30 min. The current study reports 25oC for 20 min. Suppose the method has been modified. Mention the term modified in the methodology section.
5. The culture was grown in nutrient media, and the time intervals decided was 4, 5, and 7 days concerning the individual strains. It would be interesting to see the colony's formation on plates since bacteria, fungi and actinomycetes have different morphology. Further, in the soil, the most common bacterial species found are E.coli. Under ideal conditions, the growth of this culture doubles. A 4 d interval for a bacterial culture can lead to the merge of colonies. A pictorial representation of these plates would be better to understand.
6. Mention the word “respectively” wherever necessary throughout the manuscript.
7. The word ending with “ly” in most cases throughout the manuscript changes the sentence meaning, and also, these sentences do not follow the standard writing. It is suggested that authors go through the manuscript and look for odd words with the ending “ly”. Replace these words for the readers to understand.
8. It is suggested that, in future studies, the analysis of TOC in Soil is carried out using a TOC analyzer for accurate results. The analytical techniques utilized in this study cannot provide accurate results but only approximate. Similar comments for N, P, K analysis
9. Though authors have concentrated on C cycling and CO2 emissions, It would be interesting to study the effect of temperature for other gases emitted from Soil such as methane (CH4) and nitrous oxide (N2O).

Additional comments

Authors are requested to improve the quality of English in the manuscript by using English professional service available such as Grammarly.

---

## Round 0.2 · Minor Revisions

Authors corrected the manuscript considering suggestions and comments, and it is improved. However, there some minor concerns still needs your attention, as mentioned by the reviewer. Please have a look at it and correct the manuscript further, considering those comments.

Reviewer 3 ·

Basic reporting

There are minor issues in the manuscript that have to be addressed. At present, I would like to reject the manuscript in its current form and suggest minor revisions.

Experimental design

1. Line 34 - microbial biomass N (MBP) - check if correct MBP or MBN
2. Line 56-57 - What is CP in RCP, LCP, and SCP.
3. THe manuscript includes lot of abbreviations. Authors are suggestion to include a abbreviation section before introduction section. Through this readers would understand the acronyms at ease while reading
4. Line 124 - 129. where is point (1) since only (2) and (3) is visible. Recheck and rewrite.
5. Line 162 and 219 - It is suggested that subtopics are provided for different methodology in the Determination of soil C fractions section and Examination of enzymes activity.
6. Remove line 209 - 211 - On the other hand........dilution
7. Check throughout the manuscript - Authors have mentioned the difference in the results mentioning it in fold. However, it is observed that authors have not mentioned the fold is determined compared to what. Recheck and rewrite for better understading of the readers
8. Figure representation of the methodology would be intersting for the readers. - just a suggestion
9. Authors are requested to check for grammatical errors throughout the manuscript. For example place a comma before respectively.

Validity of the findings

Authors have compared the results with reported literatures. The quality of the manuscript has improved

---

## Round 0.3 · accepted · Accept

The authors revised their manuscript as per suggestions, and it can be accepted in the current revised version.

Reviewer 3 ·

Basic reporting

Authors have carried out the the changes carefully and the manuscript quality has improved. I recommend the acceptance of the manuscript for publication in PeerJ

Experimental design

All the sugestions and comments have been carried out carefully.

Validity of the findings

All the sugestions and comments have been carried out carefully.

Additional comments

Authors have improved the the quality of English in the manuscript and have corectted the grammatical errors in the manuscript.